# Biophysical feedback of global forest fires on surface temperature

Zhihua Liu [1,2], Ashley P. Ballantyne [2] & L. Annie Cooper[2]

The biophysical feedbacks of forest fire on Earth's surface radiative budget remain uncertain at the global scale. Using satellite observations, we show that fire-induced forest loss accounts for about 15% of global forest loss, mostly in northern high latitudes. Forest fire increases surface temperature by 0.15 K (0.12 to 0.19 K) one year following fire in burned area globally. In high-latitudes, the initial positive climate-fire feedback was mainly attributed to reduced evapotranspiration and sustained for approximately 5 years. Over longer-term (> 5 years), increases in albedo dominated the surface radiative budget resulting in a net cooling effect. In tropical regions, fire had a long-term weaker warming effect mainly due to reduced evaporative cooling. Globally, biophysical feedbacks of fire-induced surface warming one year after fire are equivalent to 62% of warming due to annual fire-related $CO_2$ emissions. Our results suggest that changes in the severity and/or frequency of fire disturbance may have strong impacts on Earth's surface radiative budget and climate, especially at high latitudes.

[1] CAS Key Laboratory of Forest Ecology and Management, Institute of Applied Ecology, Chinese Academy of Sciences, Shenyang 110016, China. [2] Department of Ecosystem and Conservation Sciences, University of Montana, Missoula, MT 59812, USA. Correspondence and requests for materials should be addressed to Z.L. (email: liuzh811@126.com)

Forests provide vital climate services directly by altering the radiation budget of Earth's surface and indirectly by sequestering carbon from Earth's atmosphere[1]. However, recent changes in the frequency and intensity of forest disturbances have the potential to undermine these climate services[2–4]. Wildfire is a prevalent natural disturbance that impacts the climate services of global forests by regulating their spatial distribution, as well as the exchange of carbon, water, and energy between the land and atmosphere[1,5]. Recent concerns of potential increases of forest fire under climate change also underscore the importance of fire–climate feedbacks[6].

Many studies have investigated the spatiotemporal heterogeneity of fire regimes and their response to climate[7–12]. On the contrary, forest fires also alter the climate via biogeochemical processes, such as emissions of greenhouse gases, aerosols, and volatile organic compounds (VOCs) into the atmosphere or sequestration of carbon via post-fire regrowth of vegetation, but also through direct biophysical processes, such as changes in the absorption or redistribution of energy at Earth's surface[1,13]. To understand climatic response, many studies have also investigated how fires have affected the radiative forcing (RF) at ecosystem to regional scales using observations[14–17] or at global scales using simulation models[18–20]. However, most studies focused on RF either from albedo-induced shortwave radiation change or greenhouse gas emissions. To the best of our knowledge, no observational-based studies have quantified the amount of forest cover loss due to fire and its impact on the full surface energy balance at the global scale. Such knowledge is critical to understanding the role of fire in earth's climate system and predicting future fire–climate interactions.

Forest climate services exhibit distinct latitudinal patterns associated with different biophysical processes[21–26]. Generally, it is thought that boreal forests tend to have a net warming effect due to higher shortwave radiation absorption resulting from their relatively low albedo (α), and tropical forests tend to have a net cooling effect due to high evapotranspiration. These well-established biophysical climate feedbacks have enabled Earth system models to simulate biosphere–atmosphere interactions due to changes in forest cover, and have provided a mechanistic understanding of the climate feedbacks from observed changes in land surface properties[21,23,27,28]. However, recent satellite evidence suggests that forest cover loss is associated with a warming of Earth's surface globally[3]. Stand- to regional-scale studies have shown that forest fires have the potential to affect regional climate through changes to the energy budget[14,17,29–32]. However, these analyses did not consider fire-induced climate feedbacks between biomes, and we still have a limited understanding of how fire alters Earth's surface radiative budget and temperature at the global scale.

Here, we used spatially and temporally consistent satellite observations from Landsat and MODIS (Moderate Resolution Imaging Spectroradiometer) to assess the biophysical climate effects from fire-induced forest change at the global scale. To explore these relationships between fire-induced forest loss and Earth's surface radiative budget, we first quantified how much forest loss was caused by fire annually from 2003 to 2014? Then, we studied the Earth's land surface radiometric temperature (LST) response to fire-induced forest loss, and explored the underlying biophysical processes. We found fire-induced forest loss accounts for about 15% of global forest loss, mainly in northern high latitudes. The biophysical feedbacks of fire on surface temperature and radiative budget vary during the post-fire succession, and depend on the dynamic interaction between evapotranspiration and albedo resulted from different fire regimes among biomes.

## Results

**Fire-induced forest loss**. Fire-induced forest loss at the global scale was quantified by overlaying MODIS annual burned area (BA) and Landsat-derived annual forest loss area at 500-m resolution from 2003 to 2014 (see Methods). We considered cells as fire-induced forest loss if BA and forest loss coincided with each other in space and time (Supplementary Fig. 1). We found that mean annual fire-induced forest loss accounted for 14.8 ± 3.3% of total forest loss between 2003 and 2014, and exhibited distinct geographic and latitudinal patterns (Figs. 1a, b). The percent of fire-induced forest loss was as high as ~30% in boreal Canada and Russia, as well as in the western US due to high fire severity (Fig. 1c) and in African seasonal-dry tropical forests due to high burn rates (Supplementary Fig. 2). Fire-induced forest loss in tropics (23 S – 23 N), which are most likely deforestation fires[33,34], accounted for about 25% of global fire-induced loss and about 4% global forest loss. Trend analysis of these short-term records showed an increasing trend in percent fire-induced forest loss in Eurasia, the western US, and Australia. However, N. America showed no trend in percent fire-induced forest loss, and Africa showed a decreasing trend (Supplementary Fig. 3a). In S. America and SE Asian tropical forests, the increasing trend in Landsat-derived tree loss (Supplementary Fig. 3d) cannot be explained by a decreasing trend in BA (Supplementary Fig. 3c, also in ref. [9]), and is most likely due to human-caused deforestation activities[35,36]. Percent of high severity fire, calculated as the ratio between fire-induced forest loss (Fig. 1a) and total forest burned by fire (Supplementary Fig. 2a), was higher in boreal and western US forests, but there was no indication of any global trend in forest fire severity (Supplementary Fig. 3b), although the burn area is decreasing trend in African savannas (Supplementary Fig. 3c), inconsistent with a recent global analysis.

**Changes in surface temperatures following forest fires**. We assessed the effects of forest fires on LST using a space-for-time approach at 0.05˚ resolution from 2005 to 2014[37]. By removing potential confounding factors including background climate, vegetation type, structure and variation, as well as topography (see Methods), we calculated the impact of forest fire on LST between fire-affected pixels (fire) and non-fire-affected pixels (control) (e.g., $\Delta LST = LST_{fire} – LST_{control}$) following fire (Supplementary Fig 4). The spatial distribution of paired fire-control samples largely mirrored percent of annual BA (Supplementary Fig 5 vs Supplementary Fig 2a).

Immediately (1 year) after fire, forest fires caused a significant increase in mean annual LST ($\Delta LST = 0.153$ K, 0.120 – 0.186 K) within BAs, with a greater increase in summer ($\Delta LST = 0.218$ K, 0.151 – 0.286 K) than winter ($\Delta LST = 0.111$ K, 0.040 – 0.181 K). However, the effects of fires on LST exhibited distinct patterns across latitudes (Fig. 2 and Supplementary Fig. 6). The mean $\Delta LST$ response was most dramatic in northern high latitudes (> 45 N), where fires resulted in a strong warming effect in summer (0.664 ± 0.038 K) and a weak cooling effect in winter (−0.164 ± 0.058 K), leading to a net annual warming effect (0.198 ± 0.044 K) (Fig. 2 and Supplementary Table 1). The fire-induced warming effect in northern high latitudes may have direct impacts on climate due to its large spatial coverage (Supplementary Fig. 7). At mid-latitudes ( > 20 S and 20 N – 45 N), fires had a stronger warming effect in summer than in winter, and had an annual warming effect. At low latitudes (20 S – 20 N), forest fires had a slight but non-significant warming effect on summer ($\Delta LST = 0.036 ± 0.059$ K), winter (0.029 ± 0.062 K), and annual (0.039 ± 0.045 K) surface temperatures (Supplementary Table 1).

Post-fire trajectories in surface LST were examined to contrast the fire-induced climate feedbacks in two biomes with very

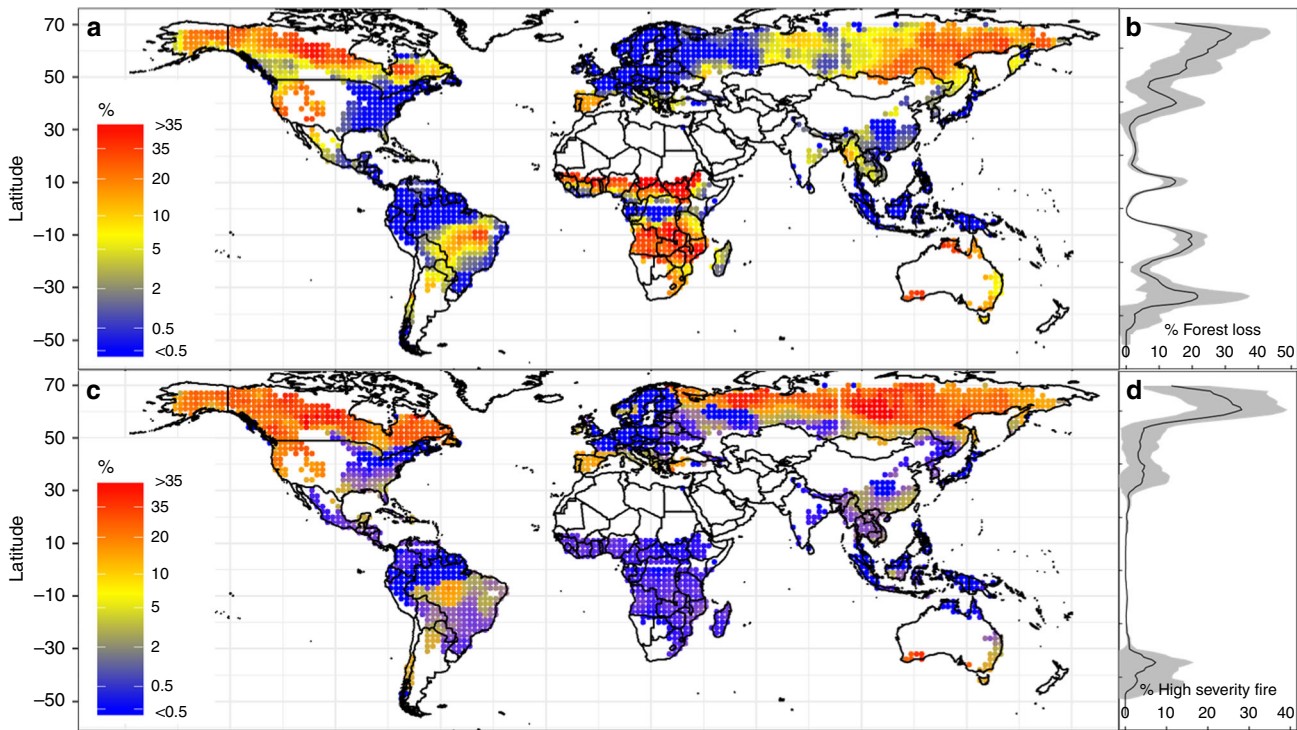

**Fig. 1** Spatial patterns of percent of mean annual fire-induced forest loss and fire severity between 2003 and 2014. Panel **a** shows the percentage of mean annual forest loss due to fire globally, and **b** binned by latitude. Panel **c** shows the percentage of high severity fire globally, and **d** binned by latitude. In **b** and **d**, shaded areas are the mean ± one standard deviation. Points are spaced 2 × 2 degree in both latitude and longitude, and smoothed by 4 × 4 degree moving windows. Panels **a**, **c** were created in the R environment for statistical computing and graphics (https://www.r-project.org/)

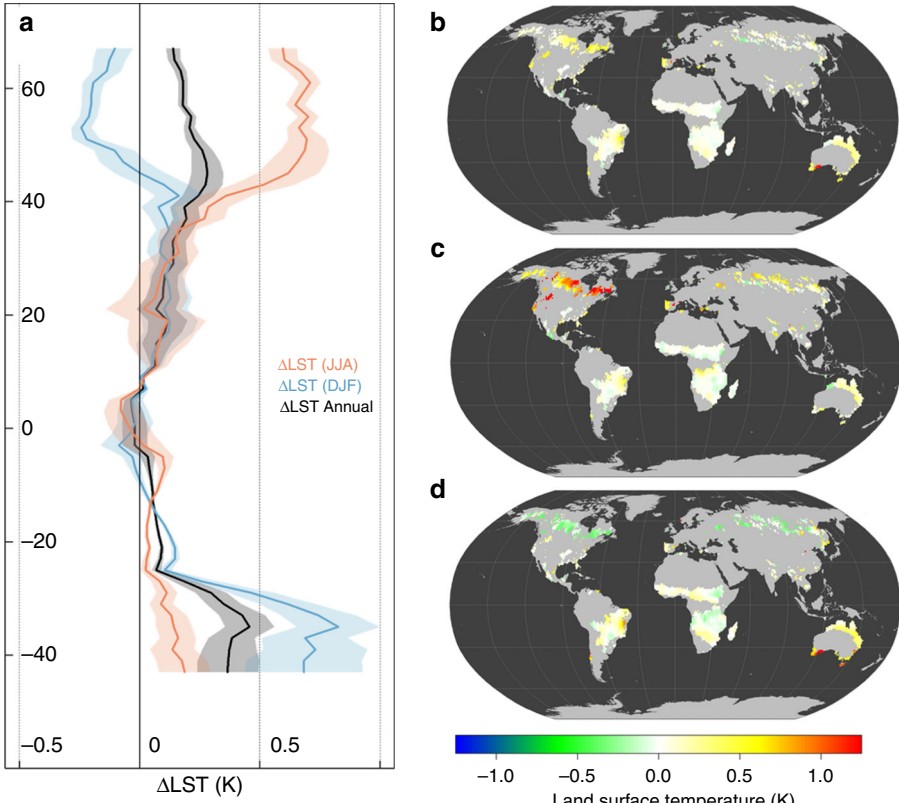

**Fig. 2** The biophysical effects of fire-induced forest loss on land surface radiometric temperature (LST). Panel **a** shows the latitudinal summary of △LST resulting from fire for annual, summer (June, July, and August (JJA)), and winter (December, January, and February (DJF)) time periods. In **a**, shaded areas are the mean ± one standard deviation. Panels **b**, **c**, **d** show spatial patterns of fire-induced △LST for annual, summer, and winter periods, respectively. Panels **b**, **c**, **d** were created in the R environment for statistical computing and graphics (https://www.r-project.org/)

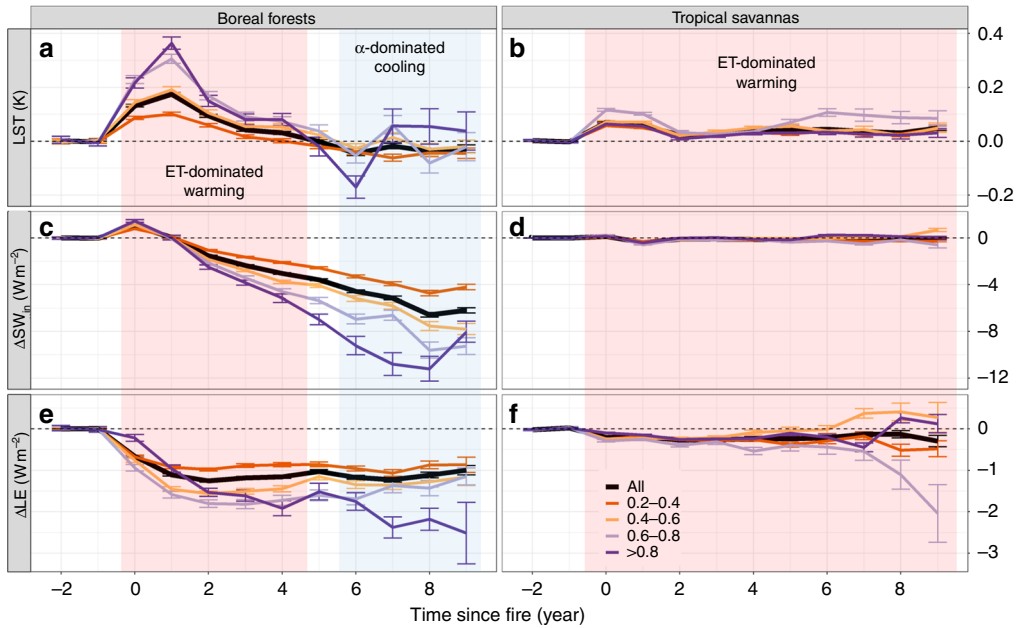

**Fig. 3** Land surface radiometric temperature (LST) after fire in boreal forests and tropical savannas. Time series plots of changes in land surface radiometric temperature ($\triangle$LST, **a, b**), shortwave radiation ($\triangle$SW$_{in}$, **c, d**), and latent heat fluxes ($\triangle$LE, **e, f**) for tropical savannas and boreal forests, stratified by different percent of area burned within each CMG grid (i.e., All, 0.2–0.4, 0.4–0.6, 0.6–0.8, > 0.8). Error bars are one standard deviation

different fire regimes and climate services (i.e., boreal forests vs. tropical savannas, Supplementary Fig. 8). Although limited by data availability (< 9 years), trajectories on the climate feedbacks of early successional stages provide good indicators on long-term trends about ecosystem dynamics and their associated biophysical processes and climate feedbacks. Several key differences were observed between boreal forests and tropical savannas in the magnitude and trend of surface temperature responses following fire events (Fig. 3). First, the magnitude and duration of the LST responses in boreal forests were more dramatic and variable than those in savannas, probably due to higher burn severity, longer ecosystem recovery time, and the seasonal presence of snow cover. Second, boreal forest fires had strong warming effects immediately after burning (< 1 year). Following these initial impacts, the magnitude of the warming effect decreased gradually to zero within 5 years following fire (Fig. 3a), followed by a net cooling effect after 5 years. As the fire return interval is usually more than several decades in boreal forests, fire may have a net cooling effect when radiative impacts are integrated over the whole fire cycle, consistent with previous findings[14]. In contrast, tropical savanna fires had a weak, but persistent, warming effect (Fig. 3b), consistent with deforestation[1].

**Changes in biophysical process following forest fire.** The underlying biophysical mechanisms behind the observed changes in surface temperature can be better understood by looking at the variations in the components of the surface energy balance (see Methods). Consistent with LST, changes in surface fluxes immediately (1 year) after fires also showed distinct latitudinal and seasonal patterns (Fig. 4). Changes in surface energy fluxes were more dramatic at high latitudes (> 45 N/30 S), possibly due to higher fire severity (Fig. 1c). At high latitudes, there was an increase in summer (June, July, August (JJA)) SW$_{in}$ absorption ($\triangle$SW$_{in}$ = 2.97 ± 0.85 W m$^{-2}$) due to a decrease in albedo ($\triangle\alpha$ = −0.0096 ± 0.0031), and a decrease in winter (December, January, February (DJF)) SW$_{in}$ absorption ($\triangle$SW$_{in}$ = −2.05 ± 0.95) due to an increase in $\alpha$ ($\triangle\alpha$ = 0.033 ± 0.0038). The combination of these changes led to a weak increase in net annual

SW$_{in}$ absorption (Fig. 3, Supplementary Figs. 9-11). The upwelling longwave radiation ($\triangle$LW$_{out}$), based on the surface energy balance, is similar to the LST response. Forest fires reduced the annual evaporative cooling ($\triangle$LE) (primarily during the growing season) due to the consumption of forest canopy and ground vegetation, together with changes in SW$_{in}$ absorption. These reductions contributed to latitudinal and seasonal changes in sensible and ground heat fluxes ($\triangle$(HE + G)). In contrast, fire-induced changes in surface energy fluxes were relatively minor and lacked seasonality at low latitudes (Fig. 4, Supplementary Figs. 9-11).

We also attributed the post-fire trajectory in surface LST to the relative importance between albedo-induced changes in SW (shortwave) radiation ($\triangle$SW$_{in}$) and ET-induced changes in LE (latent heat) fluxes ($\triangle$LE), within 9 year following fire (see Methods). The strong warming effect immediately (< 1 year) after fire in boreal forests was due to both a reduction in evaporative cooling and an increase in SW$_{in}$ absorption (Figs. 3b, c). As the vegetation reestablished, the surface radiative budget becomes dominated by increasing surface albedo thus contributing to a long-term cooling effect. Decreases in ET due to forest canopy loss leveled off after 2 years, possibly because reductions in forest ET (evapotranspiration) were partially offset by increases in grass/shrub ET. As a result, the initial net warming effect caused by decreased LE was slowly outweighed by the cooling effect caused by decreases in absorbed $\triangle$SW$_{in}$ causing a cooling trend after 5 years (Fig. 3 and Supplementary Fig. 12). In contrast, warming effects in tropical savannas were caused primarily by reduced evaporative cooling (Supplementary Fig. 12). Changes in annual albedo and subsequent effects on SW forcing were very weak, as albedo recovered within several months after fire[17], but evapotranspiration required a much longer time to recover.

The contrasts between boreal forest and tropical savanna LST responses and surface energy fluxes imply that fire regime characteristics are important for determining surface energy impacts. Therefore, we related $\triangle$LST to fire severity (% of high severity fire) and BA (% of area burned) within each 2 × 2 degree region in order to explore their impacts on the surface energy budget (see Methods). We found a stronger non-linear

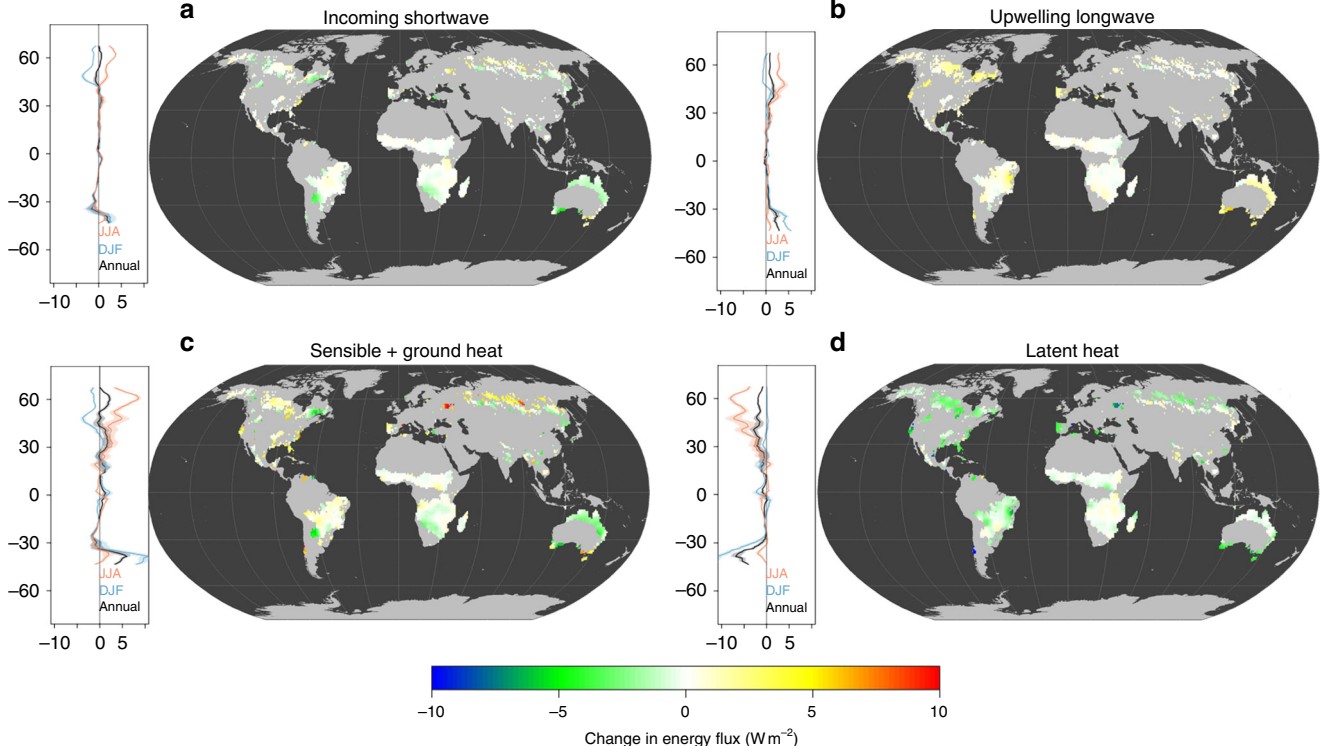

**Fig. 4** Change in energy balance after one year after fire. Immediate (1 year) impacts of fire-induced forest loss on the surface energy flux as a result of changes in **a** incoming shortwave radiation ($\triangle SW_{in}$), **b** upwelling longwave radiation ($\triangle LW_{out}$), **c** the combination of sensible and ground heat fluxes ($\triangle(HE + G)$), and **d** the latent heat flux ($\triangle LE$). For the latitudinal summaries of impacts, black lines represent annual impacts, red lines represent summer (June, July, August; JJA) impacts, and blue lines represent winter (December, January, February; DJF) impacts. The shaded areas are the mean ± one standard deviation. Panels **a**, **b**, **c**, **d** were created in the R environment for statistical computing and graphics (https://www.r-project.org/)

relationship between fire severity and summer LST ($R^2 = 0.63$) than between BA and summer LST ($R^2 = 0.02$), suggesting that fire severity may have a bigger influence on perturbations to the surface radiative budget than BA (Fig. 5 and Supplementary Fig. 13), which is consistent with previous findings at the regional scale[30,38,39]. We also related $\triangle LST$ to BA and fire severity at the 0.05° scale, which showed a stronger response of $\triangle LST$ to fire severity, especially in tropical regions (Supplementary Fig. 14). Among biomes, tropical fires are characterized by higher burn frequency but lower fire severity, which partially explains the dampened response of $\triangle\alpha$, $\triangle ET$, and $\triangle LST$ within tropical regions. In contrast, boreal forest fires are typically higher in severity but lesser in areal extent, leading to stronger responses in surface biophysical properties (e.g., albedo, forest canopy consumption) and resultant climate feedbacks and energy flux perturbations. The non-linear relationship between $\triangle LST$ and fire severity (Fig. 5) suggests that small changes in fire severity in typically low fire severity regions, such as the tropics and Siberian boreal forests, may result in potentially large changes in the post-fire LST response.

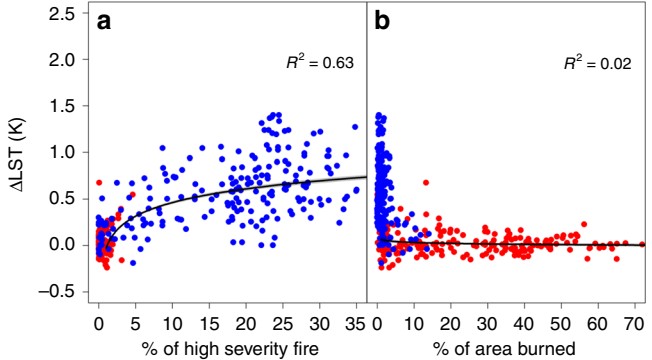

**Fig. 5** Effects of fire severity and burned area on summer (June, July, August; JJA) temperature. Panels show the relationship between land surface temperature change and **a** the percentage of high severity fire, and **b** the percentage of burned area. Blue points show results from boreal forests, whereas red points show results from the tropical savannas within each 2 × 2 degree region

## Discussion
One of the main sources of uncertainty for climate predictions is the response of terrestrial ecosystems, and land surface change due to disturbance has been found to be an important factor contributing to that uncertainty[40]. Our results show that immediate biophysical climate impacts from fire-induced forest loss may be as strong as those associated with human-induced land cover change. During the study period, we calculated that the global mean surface warming 1 year after forest fire was 0.0046 K (95% confidence interval (CI) = 0.0042 – 0.0051 K, see Methods), which is comparable to the rate of long-term global

surface warming at 0.0064 K yr⁻¹ between 1880 and 2012[41,42]. The immediate surface warming effect is also of roughly the same magnitude as the effects of total global forest cover change (e.g, 0.0062 K)[3], irrigation, and land cover change[43]. Recent estimates have shown that biomass burning results in carbon emissions of 2.2 Pg C yr⁻¹, almost double the 1.3 ± 0.7 Pg C yr⁻¹ emitted due to human-induced land cover change[44], and equivalent to about 20% of fossil fuel carbon emissions (~9.4 ± 0.5 Pg C yr⁻²)[44]. Of total carbon emissions from biomass burning, roughly 0.55 Pg C yr⁻¹ was human related, including agriculture waste burning, tropical deforestation, and peatland fire[45,46], although recent

estimates have shown global fire activity has reduced by 25% due to agriculture activity in tropical region for the last two decades[9]. However, globally, wildfire disturbs more forests (includes both lethal and non-lethal fires) than human-caused forest cover change, especially in the boreal region[47]. Our results suggest that the immediate surface RF following forest fire is equivalent to 61% of the biogeochemical atmospheric RF due to annual $CO_2$ emissions from biomass burning (see Methods).

The contrasting results from boreal forests and tropical savannas indicate that climate–fire feedbacks are biome specific and temporally variable. Specifically, these results suggest variation relating to background climate, forest species composition, and fire regime may have an impact on the biophysical response of ecosystems to fire. We found that fires had an initial strong positive climate feedback in boreal forests primarily due to warming effects from reduced summer evaporative cooling resulting from canopy loss. At longer time scales (> 5 years), boreal forest fires had a strong cooling effect, mainly due to increased $SW_{in}$ absorption due to higher albedo. While this albedo response is consistent with results from several other studies investigating North American boreal forests[14,31,32,48], our findings indicate that declines in evapotranspiration are necessary to explain the observed land surface temperature response. In contrast with boreal fire effects, fires had a persistent warming effect in tropical savannas due to reduced evaporative cooling, consistent with the expectation that the loss of tropical forests should lead to a surface warming[1]. Although, we found albedo-induced changes in shortwave RF were very weak at the annual scale in tropical savannas, albedo-induced changes have been found to be more important on seasonal time scales[17]. Furthermore, patterns of land-use change following fire may vary considerably in different regions of the tropics and may determine which processes come to dominate the surface radiative budget over time[49]. Most previous studies have studied the climate feedbacks resulting from changes to albedo-induced shortwave radiative absorption or RF caused by greenhouse gas emissions. However, our results suggest that reduced evaporative cooling is another key mechanism in regulating climate–fire feedbacks, and is a major driver of short-term positive climate–fire feedbacks from the biome to global scales.

Our results also indicated that fire-induced forest loss may alter Earth's surface radiative budget through different biophysical processes, despite the limited satellite observations. Specifically, short-term fire-induced changes in albedo exhibited distinct seasonal patterns, which appeared to cancel each other out between snow and snow-free seasons (Supplementary Figs. 8-9 and Supplementary Table 1), therefore leaving decreased evapotranspiration during the growing season as the dominant control on the surface radiative budget and thus temperature response in early post-fire successional stages (e.g., < 5 years in boreal forests). Over the long-term, however, shortwave radiative change due to fire-induced albedo increase dominates the surface radiative budget potentially leading to negative climate feedback, which is consistent with observationally based studies at the ecosystem, regional, and global scales studies[14,20,32]. Our results are also consistent with a recent analysis indicating that forest loss globally resulted in a slight increase in LST[3] across all biomes, although our results also suggest that most of this temperature response is due to reduced evapotranspiration in the case of fire. Recognition that surface climate feedbacks vary by different types and severities of land cover change may help to reconcile apparently inconsistent results from previous observational studies on forest loss[3] and land cover change, as well as model simulations of the same processes. Consequently, understanding the climate feedback from various types of forest disturbances is necessary to understand the climate feedbacks due to changes in global forest cover.

Our results suggest that fire severity is a much better predictor of changes in land surface radiative budget than BA. Thus, stronger climate–fire feedbacks may be expected in high latitude ecosystems where fire regimes are predicted to become more frequent and severe[50–57]. Although it has been shown that BA may be declining globally, our results do not necessarily show any significant change in burn severity. Therefore, even though the areal extent of fire may be declining, the biophysical and biogeochemical effects of fire may still be affecting Earth's radiative budget. If an intensified boreal fire regime unfolds as predicted, the magnitude and duration of the initial positive climate–fire feedbacks may be greatly enhanced, and this may have stronger influence on Earth's surface radiative budget. Multiple lines of evidence, including increases in BA, severity, and carbon emissions, forest species changes and expansion, and permafrost thaw, have suggested that biogeochemical and biophysical properties are rapidly changing in high latitude ecosystems. Therefore, understanding the vulnerability of high latitude ecosystems in response to changing climate–fire feedbacks remains critical.

Despite the importance of fire globally, the simulation of fire in Earth system models varies greatly in how spatial distributions, trends, and carbon emissions are implemented, and simulations are not necessarily consistent with observations[9]. Moreover, current efforts focus on characterizing BA and the biogeochemical processes affecting carbon emissions and tend to have overly simplified representations of fire severity and biophysical climate feedbacks[18]. Additionally, most current models do not accurately characterize post-fire succession and its biophysical feedbacks on energy fluxes[58]. Here, we present a framework for assessing the surface radiative impacts due to fire, which can be applied to a range of land surface disturbance types. The biophysical diagnostics presented here are also prognostic variables within land surface models that can help identify what processes the models are accurately simulating and what processes need improvement to increase our confidence in future climate predictions.

## Methods

**Spatial overlay approach to quantify global fire-induced forest loss**. We overlaid MODIS BA and Landsat-derived forest loss at 500 m resolution over forested areas (tree cover > 20% in 2000 based on[35]) for each year from 2003 to 2014. To reduce potential uncertainties, we only analyzed the pixels with > 20% tree cover in 2000, and also limited the study to pixels in which a maximum of one fire per year had occurred. Pixels were labeled as having experienced fire-induced forest loss if the MODIS BA data coincided with Landsat-derived forest loss for the fire year and 2 years postfire (i.e., $t + 0$, $t + 1$, $t + 2$). This approach was used to account for delayed tree mortality after fires. Following identification of fire-induced forest loss pixels, we summarized the total forest area ($Area_{forest}$ > 20% tree cover in 2000), total BA based on MODIS BA data ($Area_{fire}$), total forest loss based on Landsat-derived annual tree loss data ($TreeLoss_{total}$), and fire-induced tree loss ($= TreeLoss_{fire}$) within each $2 \times 2$ degree region. We derived four variables for each $2 \times 2$ degree region (Supplementary Fig. 1): variable 1, defined as the percent of total forest loss = $TreeLoss_{total}/Area_{forest}$. Variable 2, defined as the percent of fire-induced forest loss = $TreeLoss_{fire}/TreeLoss_{total}$. Variable 3, defined as the percent of forest burned by fire = $Area_{fire}/Area_{forest}$. Variable 4, defined as the percent of fire that is high severity = $TreeLoss_{fire}/Area_{fire}$.

Finally, these four variables were plotted at $2 \times 2$ degree spatial resolution, and smoothed over $4 \times 4$ degree moving windows (Fig. 1, Supplementary Fig. 2). Trends for these four variables were assessed using linear regression, and the significance level was set at $\alpha = 0.1$. Trends were plotted as Supplementary Fig. 3.

**Space-for-time approach**. We used a space-for-time approach, in which we compared the LST difference ($\Delta LST$) between burned pixels ($LST_{fire}$) and adjacent unburned pixels ($LST_{control}$) to assess the climate effects of fire on local surface temperatures at 0.05° spatial resolution:

$$\Delta LST = LST_{fire} - LST_{control} \qquad (1)$$

Positive (negative) $\Delta LST$ indicates a warming (cooling) effect due to fire. Differences in albedo ($\Delta\alpha$), ET ($\Delta ET$), and energy fluxes (e.g., incoming shortwave radiation ($\Delta SW_{in}$), upwelling longwave radiation ($\Delta LW_{out}$), latent heat fluxes ($\Delta LE$), and sensible and ground heat fluxes $\Delta(HE + G)$) are defined similarly[21].

The key goal for this space-for-time approach was to make sure that the change in land surface temperature and energy fluxes was only due to fire, and not due to other factors such as background climate conditions, vegetation type, structure, and dynamics, or the physical environment. To achieve this goal, we employed a three-step procedure (Supplementary Fig. 4).

In the first step, for each fire pixel, we overlaid a $9 \times 5$ pixel search window (longitude × latitude, approximately equal to 50 km × 28 km). Only unburned pixels within the window with the same background climate (defined by Köppen–Geiger climate region map: Supplementary Fig. 8) and forest type (defined by MODIS land cover data: Supplementary Fig. 15) as the fire pixel were selected as candidate control pixels. This step ensured that the fire and control pixels were within the same background climate and forest type.

In the second step, time series of spectral indices, including NDVI (Normalized Difference Vegetation Index), LST, and albedo ($\alpha$), were compared between fire pixels and each candidate control pixel for 2 years before the fire year ($t - 2$, $t - 1$, and $t$). If the difference in spectral indices between fire and candidate control pixels was smaller than a threshold ($\Delta$NDVI < 0.05, $\Delta$LST < 0.2, $\Delta\alpha$ < 0.05), the candidate control pixels were retained. After this procedure, post-fire ($t + 1$) spectral indices for each candidate control were checked relative to prefire time series ($t - 2$, $t - 1$, and $t$) of the same pixel in order to ensure consistency within the control pixels. This step made sure that fire and control pixels were stable and comparable in vegetation characteristics and biophysical parameters.

In the third step, the difference in physical environment (i.e., elevation, aspect, and slope) between fire and control pixels were also compared. If the physical environment between fire and candidate control pixels were similar ($\Delta$elevation < 50 m, $\Delta$slope < 5, $\Delta$aspect < 30), the control pixels were retained. This step ensured that fire and control pixels were topographically similar enough for comparison. Only the control pixel located nearest to the fire pixel was used if more than one candidate control pixel was found following the above-mentioned procedure.

A total of 92,452 paired fire-control samples were identified between 2005 and 2013 with higher numbers in the tropics and lower numbers in boreal regions (Supplementary Fig. 5).

**Calculating changes in surface energy fluxes**. The full surface energy balance of the land surface can be expressed as:

$$R_{net} = (SW_{in} - SW_{out}) + (LW_{in} - LW_{out}) = (1 - \alpha)SW_{in} + \sigma\left(\varepsilon_{cloud}T_{cloud}^4 - \varepsilon_{surf}LST_{surf}^4\right) \tag{2}$$

$$R_{net} = HE + LE + G \tag{3}$$

Based on Stefan–Boltzmann's law, net radiation ($R_{net}$) is the balance between the inputs and outputs of shortwave ($SW_{in}$, $SW_{out}$) and longwave radiation ($LW_{in}$, $LW_{out}$), and is primarily controlled by surface albedo ($\alpha$), emissivity and the temperature of clouds ($\varepsilon_{cloud}$, $T_{cloud}$) and the land surface ($\varepsilon_{surf}$, $LST_{surf}$), and the Stefan–Boltzman constant ($\sigma$, $5.67 \times 10^{-8}$ W m$^{-2}$ K$^{-4}$). Then, the net radiative energy absorbed by an ecosystem (e.g., $R_{net}$) or land surface, is approximately balanced by energy that is transferred out of the ecosystem by non-radiative processes, including LE (latent heat flux), HE (sensible heat flux), and G (ground heat flux)[59].

For the specific goals of this analysis, we are interested in how the terms of this equation change as a result of fire-induced forest loss. Following Duveiller et al.[2], we make the assumption that fire-induced forest loss is too small to generate strong feedbacks in the cloud regime, and as a consequence we assume $\Delta\varepsilon_{cloud} = 0$ and $\Delta T_{cloud} = 0$ (i.e., no change in cloud emissivity and temperature). Therefore, the change in sensible and ground heat fluxes ($\Delta(H + G)$ under clear sky conditions can be calculated by rearranging Eqs. 1 and 2:

$$\Delta(HE + G) = (1 - \Delta\alpha)SW_{in} + \sigma\left(-\Delta\varepsilon_{surf}\Delta LST_{surf}^4\right) - \Delta LE \tag{4}$$

Therefore, the change in absorbed incoming shortwave radiation on the land surface ($\Delta SW_{in}$) can be calculated from changes in albedo and incoming shortwave radiation (i.e., $\Delta SW_{in} = 1 - \Delta\alpha)SW_{in}$, the latter being available from CERES data at 1° resolution.

Changes in upwelling longwave radiation by the land surface ($\Delta LW_{out}$) can be approximated from changes in mean surface radiometric temperatures ($LST_{surf}$), which are the average of day-time and night-time LST from the MYD11C3 product, and broadband emissivity ($\Delta\varepsilon_{surf}$) (i.e., $\Delta LW_{out} = -\sigma \Delta\varepsilon_{surf}\Delta LST_{surf}^4$)[4]. Broadband emissivity ($\varepsilon_{surf}$) can be calculated from an empirical equation[60] as follows:

$$\varepsilon_{surf} = \varepsilon_{29} + \varepsilon_{31} + \varepsilon_{32} \tag{5}$$

where $\varepsilon_{29}$, $\varepsilon_{31}$, $\varepsilon_{32}$ are the estimated emissivity in MYD11C3 product bands 29 (8400 – 8700 nm), 31 (10,780 – 11,280 nm), and 32 (11,770 – 12,270 nm).

$\Delta LE$ can be calculated using the difference in evapotranspiration ($\Delta ET$) between fire and control pixels ($\Delta LE = \Delta ET \times 28.94$ W m$^{-2}$/mm day$^{-1}$, ET in mm day$^{-1}$).

**Long-term post-fire changes**. Changes in land surface temperatures ($\Delta LST$) and energy fluxes (i.e., $\Delta SW_{in}$, $\Delta LE$) up to 9 years post fire were contrasted between boreal forests and tropical savannas due to their identified importance in determining the global climate feedback from fires. To remove the potential confounding effect of fire-induced vegetation change, the land cover type was examined between prefire (up to 2 years before fire) and post-fire (up to 9 years after fire) periods. Pairs of fire-control were used to plot long-term post-fire dynamic only if there is no land cover change observed by MODIS. To evaluate the effect of BA on climate feedbacks, post-fire trajectories in $\Delta LST$, $\Delta SW_{in}$, and $\Delta LE$ were also stratified by percent BA (PBA) within 0.05° grid cells (i.e., All, 0.2–0.4, 0.4–0.6, 0.6–0.8, > 0.8).

To compare the potential response of land surface radiometric temperature ($\Delta LST$) to different components of the fire regime (i.e., fire severity and BA), we examined the correlations between $\Delta LST$ (Fig. 2) and the percent of high severity fire (Fig. 1c), as well as the PBA (Supplementary Fig. 1c) over $2 \times 2$ degree windows for summers and winters, and annually (Fig. 5).

**Estimation of warming due to biomass burning emissions**. Wildfire impacts climate by changing the atmospheric concentration of greenhouse gases (biogeochemical effect; $CO_2$, $CH_4$, $N_2O$), aerosol–radiation interactions, and ozone concentrations. Based on the IPCC AR5 report, the net RF of biomass burning emissions for aerosol–radiation interactions is close to zero, but with strong positive RF from black carbon and negative RF from organic carbon emissions[61,62]. Wildfire generally has a positive effect on RF through effects on ozone concentrations, but no reliable estimate exists of this effect[13]. Here we focus on the biogeochemical climate effects due to the annual release of $CO_2$ through biomass burning.

Based on the GFED4.1s, C emissions from biomass burning between 1997 and 2016 are estimated at $1.99 \pm 0.29$ PgC yr$^{-1}$, equivalent to $7.3 \pm 1.06$ pG $CO_2$ yr$^{-1}$ [45]. Assuming an airborne fraction of biogenic $CO_2$ fluxes of 43%[63] and considering that 1 ppm of atmospheric $CO_2$ corresponds to 7.82 Gt $CO_2$, wildfire emissions led to an increase of $0.402 \pm 0.059$ ( $= 7.3 \times 0.43/7.82$) ppm $CO_2$ yr$^{-1}$ in the atmosphere.

The RF and related climate warming induced by this change in atmospheric composition can be approximated with the following equation, using the intermediate year in the series (2007) as the base year for the calculation of the $CO_2$ mixing ratio:[64]

$$RF = 5.35\ln((383.76 + 0.402)/383.76) \tag{6}$$

Assuming a transient climate response of 1.33 K/(W m$^{-2}$)[65], the RF resulting from the emissions resulting from biomass burning ($0.0056 \pm 0.00082$ W m$^{-2}$) has led to a warming of global air surface temperatures of $0.0075 \pm 0.0011$ K yr$^{-1}$.

To isolate the contribution of the biophysical climate impacts of wildfires for each climate region, we computed the area-averaged local effects shown in Fig. 2d for each climate region, Contribution$_i = \Delta LST_i \times Area_i / Area_{global}$ where $\Delta LST_i$ ($i = 1, 2, 3, 4$, for equatorial, arid, temperate, and boreal regions) is the averaged local effect of each climate region, Area$_i$ is the area of each climate region, and Area$_{global}$ is the area of the global land surface, excluding Antarctica.

Following this methodology, we computed the mean biophysical climate impacts of wildfire to be 0.0164 K (95% CI = 0.0149 – 0.0179 K), with the contributions of each climate region as follows: (1) boreal, 0.0045 K (95% CI = 0.00396 – 0.00507 K), (2) temperate: 0.003 K (95% CI = 0.0021 – 0.0038 K), (3) arid: 0.003 K (95% CI = 0.00257 – 0.00418 K), and (4) equatorial: 0.0034 K (95% CI = 0.00257 – 0.00417 K).

In order to estimate the relative importance of the biogeochemical versus biophysical climate impacts of wildfire, we compared the rates of biogeochemical warming with the spatial average of the biophysical effects over the global land + ocean area observed between 2005 and 2014. We computed the biophysical climate effects over the globe to be 0.0046 K (95% CI = 0.0042 – 0.0051 K), equivalent to 61% of the biogeochemical effects of biomass $CO_2$ emissions, as calculated above.

**Köppen–Geiger climate region map**. Climate regions were derived from Köppen–Geiger world map at 0.5° spatial resolution, representing the period 1951–2000. The map contains 31 climate zones (http://koeppen-geiger.vu-wien.ac.at/present.htm), and was reclassified into five major regions (equatorial, arid, temperate, boreal, and polar)[66]. The polar zone was excluded because it does not include forested areas and therefore forest fires (Supplementary Fig. 8).

**Annual forest loss data**. Annual forest loss data are available at 30 m resolution from 2000 to 2014[35]. High-resolution forest loss data were aggregated into 500 m resolution, so as to be consistent with the MODIS BA product (MCD64A1) and allowing for the calculation of annual burned forest area and fire-induced tree mortality. The tree cover in 2000 was contained in the forest cover change dataset and was used to define forested area. The forest loss data used in the analysis are available online (http://earthenginepartners.appspot.com/science-2013-global-forest/download_v1.4.html).

**Land surface radiometric temperature**. We used LST data from the MYD11C3 (V6) AQUA product at 0.05° resolution and a monthly time step. This product was chosen because it better approximated the timing of surface retrievals of maximum

and minimum temperatures than the MOD11C3 TERRA product. We used maximum LST (day-time LST at approximately 1:30 p.m. local time), minimum LST (night-time LST at approximately 1:30 a.m. local time), and mean LST (average of maximum and minimum LST) for this analysis. MODIS LST (V6) has been widely validated against a variety of land cover types and generally shows good correlations with ground values[67]. Emissivity values in this product are estimated based on land cover type and therefore are not likely to accurately change following low severity fires. However, potential emissivity values cover a small range (0.95–0.98) and therefore have a relatively small influence on LST. To reduce uncertainties associated with emissivity values, we only used high quality LST values (emissivity error $<= 0.02$ and LST error $<= 1$ K).

**Normalized difference vegetation index**. We used NDVI data from the MYD13C2 (V6) product at 0.05° resolution and a monthly time step. Fire severity as shown in Supplementary Fig. 14 was calculated as the NDVI change one year after fire (ΔNDVI) between burned pixels ($NDVI_{fire}$) and adjacent unburned pixels ($NDVI_{control}$) at 0.05° spatial resolution ($\Delta NDVI = NDVI_{fire} - NDVI_{control}$). Only pixels in which QA was flagged as 0 (good data) and 1 (marginal data) were used.

**Albedo**. We used white-sky shortwave albedo (α) from the MCD43C3 (V5) product at 0.05° and a 16-day-time step. The white-sky albedo product is derived from bidirectional reflectance distribution function (BRDF) measurements, integrated over both incoming and outgoing hemispheres, and does not depend on illumination or atmospheric conditions. Black-sky albedo (directional hemispherical reflectance) is the albedo in the absence of a diffuse component and is a function of solar zenith angle. Actual albedo uses both black- and white-sky albedo weighted by the proportion of direct and diffuse illumination, which is not currently available. We chose to use white-sky albedo, rather than black-sky or actual (blue-sky) albedo, because previous studies have found the choice of specific albedo (white-, black-, or blue-sky) has little impact on analyses at the global scale?[21,68,69] and black-sky and white-sky albedo is highly correlated[70]. Pixels in which the QC was flagged as 0 (best quality), 1 (good quality), or 2 (mixed quality) were used. The 16-day albedo data were aggregated to a monthly time step based on the date indicated in the metadata.

**Evapotranspiration**. We used MOD16 ET data at 0.05° resolution and a monthly time step. The MOD16 ET algorithm is based on the Penman–Monteith equation[71]. ET data were downloaded from the NTSG website (http://www.ntsg.umt.edu).

**Land cover**. Land cover data were based on the 0.5 km resolution MODIS-based Global Land Cover Climatology (2001–2010) product using the IGBP classification[72]. We defined forest extent as any pixel in which tree cover was >20%, thereby corresponding to the forest and savanna land cover types based from MODIS (Supplementary Fig. 15).

**BA**. We used BA data from the MCD64A1 BA product, available at 500 m resolution and a monthly time step[73]. In this analysis, monthly pixel-level BA was first integrated into annual pixel-level BA (1-burned, 0-unburned), and then aggregated into PBA at 0.05° spatial resolution (# of burned pixels divided by # of pixels within the 0.05° grid). To reduce potential uncertainties resulting from MODIS observations, only PBA >0.2 were included in the analysis. PBA at 0.05° spatial resolution was used to select the paired fire-control samples for the assessment of the biophysical climate effects of fire and its controls.

**Radiation**. Monthly gridded incoming shortwave ($SW_{in}$) solar radiation at 1° spatial resolution from 2003 to 2015 was used to calculate the net surface radiation change due to changes in albedo. SW↓ is based on global surface radiative flux data —CERES EBAF-Surface v4.0—which are available from the NASA Clouds and the Earth's Radiant Energy System (CERES)[74] (https://ceres.larc.nasa.gov/index.php).

## Data availability

All data analyzed in this study are publicly available. All MODIS data are available from the Land Processes Distributed Active Archive Center (LP DAAC: https://lpdaac.usgs.gov/). ET data are available from the NTSG group at the University of Montana (http://www.ntsg.umt.edu/). The Köppen–Geiger World climate zone map is available online (http://koeppen-geiger.vu-wien.ac.at/present.htm). Radiation data are available from the NASA Clouds and the Earth's Radiant Energy System (CERES) site (https://ceres.larc.nasa.gov/index.php). Forest loss data are also available online (http://earthenginepartners.appspot.com/science-2013-global-forest/download_v1.4.html).

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

## Acknowledgements

Z.L. was supported by NSF grant (1550932), NSFC (31470517), CAS Pioneer Hundred Talents Program, and K.C.Wong Education Foundation. L.A.C. was supported by NASA Earth and Space Science Fellowship (#NNX15AN16H). The authors thank Professor Steven W. Running for providing comments on the manuscript.

## Author contributions

Z.L. conceived and designed the study, conducted the analysis, and led the manuscript writing. A.P.B. and L.A.C. contributed to results interpretation and manuscript writing.

## Additional information

**Competing interests:** The authors declare no competing interests.

