## [Peer Review File · Nature Communications]

Editorial Note: This manuscript has been previously reviewed at another journal that is not operating a transparent peer review scheme. The manuscript was considered suitable for publication without further review at *Nature Communications*.

Reviewer #1 (Remarks to the Author):

I reviewed a previous version of this manuscript and found it to be a valuable contribution to the literature and scientific understanding of fire-climate interactions, and based on sound methodology. At the time, however, I offered some criticisms regarding how the authors framed and contextualized their work which needed to be addressed before publication.

I commend the authors for addressing these criticisms, and I now believe the broader view of how this research fits into the literature is appropriate. Some minor comments are offered below:

-L 16. Add 'K' after 0.15

-L 16. I would delete the word 'global', or move it to be the first word in this sentence. As is, the language is confusing and implies the authors are assessing global mean surface temperature (ie both burned and unburned pixels)

-L23: change 'is' to 'are'

-L27. Might also mention the impact of forest-emitted VOCs (e.g., isoprene).

-L 57: I would change 'among' to 'between'

-Fig 1, S2: why is there an apparent missing line of data in central/western Siberia?

-L82: I would not say this is 'inconsistent' since the variables are different and explained, as the authors say, most likely by deforestation

-L121: misspelled 'several'

-L135. remove ' _ '

-L187. change to plural carbon emissions

-L188. change to plural estimates

Response to reviewers

REVIEWERS' COMMENTS:

Reviewer #1 (Remarks to the Author):

I reviewed a previous version of this manuscript and found it to be a valuable contribution to the literature and scientific understanding of fire-climate interactions, and based on sound methodology. At the time, however, I offered some criticisms regarding how the authors framed and contextualized their work which needed to be addressed before publication.

[Comments 1.1] I commend the authors for addressing these criticisms, and I now believe the broader view of how this research fits into the literature is appropriate. Some minor comments are offered below:

[Response to Comments 1.1] Thank you for your suggestions and comments on previous version of this manuscript. They are helpful in guiding our revision, and greatly improve the quality of our work.

[Comments 1.2]

-L 16. Add 'K' after 0.15

-L 16. I would delete the word 'global', or move it to be the first word in this sentence. As is, the language is confusing and implies the authors are assessing global mean surface temperature (ie both burned and unburned pixels)

-L23: change 'is' to 'are'

-L27. Might also mention the impact of forest-emitted VOCs (e.g., isoprene).

-L 57: I would change 'among' to 'between'

[Response to Comments 1.2] These minor comments and editorial suggestions has been addressed. (line 14, 15, 20, 35-36, 57)

[Comments 1.3]-Fig 1, S2: why is there an apparent missing line of data in central/western Siberia?

[Response to Comments 1.3] We estimated the value region by region, and there is a 0.5 degree gap in that regions due to region delineation scheme. However, this should not affect our results.

Response to reviewers

[Comments 1.4]-L82: I would not say this is ‘inconsistent’ since the variables are different and explained, as the authors say, most likely by deforestation

[Response to Comments 1.4] we have changed this into “cannot be explained by” (line 86).

[Comments 1.5]

-L121: misspelled ‘several’

-L135. remove ‘_’

-L187. change to plural carbon emissions

-L188. change to plural estimates

[Response to Comments 1.5] These editorial suggestions has been adopted.